# Factor XIa Inhibitors as a Novel Anticoagulation Target: Recent Clinical Research Advances

**DOI:** 10.3390/ph16060866

**Published:** 2023-06-11

**Authors:** Yunqing Xia, Yu Hu, Liang Tang

**Affiliations:** 1Institute of Hematology, Union Hospital, Tongji Medical College, Huazhong University of Science and Technology, No. 1277 Jiefang Avenue, Wuhan 430022, China; m202175806@hust.edu.cn; 2Collaborative Innovation Center of Hematology, Huazhong University of Science and Technology, Wuhan 430022, China

**Keywords:** XIa, venous thromboembolism, clinical trials

## Abstract

Background: While current clinically administered anticoagulant medications have demonstrated effectiveness, they have also precipitated significant risks: severe bleeding complications including, but not limited to, gastrointestinal hemorrhaging and intracranial and other life-threatening major bleedings. An ongoing effort is being made to identify the best targets for anticoagulant-targeted drugs. Coagulation factor XIa (FXIa) is emerging as an important target of current anticoagulant treatment. Objective: This review will summarize the development of anticoagulants and recent advances in clinical trials of experimental factor XI inhibitors from a clinical application perspective. Results: As of 1 January 2023, our search screening included 33 clinical trials. We summarized the research progress of FXIa inhibitors from seven clinical trials that evaluated their efficacy and safety. The results showed no statistically meaningful distinction in the primary efficacy between patients receiving FXIa inhibitors compared to controls (RR = 0.796; 95% CI: 0.606–1.046; I^2^ = 68%). The outcomes did not indicate a statistical difference in the occurrence of any bleeding between patients receiving FXIa inhibitors compared to controls (RR = 0.717; 95% CI: 0.502–1.023; I^2^ = 60%). A subgroup analysis found significant differences in severe bleeding and clinically relevant hemorrhaging in subjects receiving FXIa inhibitors compared to Enoxaparin (RR = 0.457; 95% CI: 0.256–0.816; I^2^ = 0%). Conclusions: Clinical trials to date have indicated that factor XIa is a potential anticoagulation target, and factor XIa inhibitors may play an important role in the development of anticoagulants.

## 1. Background

Thromboembolic disease, including venous and arterial thrombosis, is one of the major causes of global mortality and handicap [1]. Venous thromboembolism (VTE) comprises pulmonary embolism (PE) and deep vein thrombosis (DVT). Arterial thrombosis primarily concerns ischemic stroke (IS) and ischemic heart disease (IHD). Atrial fibrillation and disseminated intravascular coagulation are associated with thromboembolism. Thrombogenesis accounts for most general latent pathological mechanism in three major cardiovascular diseases: IHD (acute coronary syndrome), IS and VTE [2,3]. Anticoagulants have long been an important cornerstone in the preventive and therapeutic measures for thrombotic illnesses. The anticoagulants used for most of the 20th century were vitamin K antagonists (VKAs) and heparin [4,5]. However, heparin and warfarin are associated with a number of drug-specific adverse reactions as well as inter-patient response differences and demands for laboratory dose adjustments. Changes in patient responses to therapy might be caused by the non-specific binding of heparin or changes in hepatic metabolism from warfarin [6]. With the development of high-throughput screening and drug design based on structure, the advance of novel oral anticoagulants (NOACs) for single targets continues at an accelerated pace [7]. Oral anticoagulants currently used in the prophylaxis and therapy of thromboembolic disorders include Apixaban, Rivaroxaban, Edoxaban, and Dabigatran, all of which inhibit proteinase Xa (FXa) and thrombin or reduce its precursors: factor X and plasma concentrations of thrombin [8]. As factor Xa and thrombin are the center in the hemostatic response to damage, these drugs increase the possibility of serious hemorrhaging in patients, particularly gastrointestinal, intracranial and life-threatening major bleeding [9]. Results of a meta-analysis of NOACs in patients with myocardial infarction suggest a large heterogeneity in the cardiovascular safety of NOACs [10]. According to studies, the annual occurrence rate of massive hemorrhaging in atrial fibrillation patients using NOACs is 2–3%, and the incidence rate of annual intracranial hemorrhaging is 0.3–0.5% [11,12]. There is an ongoing search for the best targets for anticoagulation drugs. Although these targets have less of a role in hemostatic response compared to the core targets, they are clinically safer from a therapeutic point of view because they reduce the side effects of hemorrhages. Overall, the ultimate target of anticoagulation is to reduce thrombogenesis without compromising hemostatic procedure. Clinical research using FXI antisense oligonucleotides (ASO) confirmed this connection by showing that reduced FXI levels led to superior prophylaxis of surgery-induced VTE and decreased hemorrhaging compared to standard anticoagulant medication [13]. Patients with FXI deficiencies seldom have spontaneous bleeding, and plasma FXI levels have no impact on the probability of bleeding [14]. Most patients with FXI insufficiency do not show any symptoms, and their main clinical symptom is an increased propensity for serious damage or a light hemorrhage following an operation [15,16,17]. A sizable case-control research based on the population found that patients who have deep vein thrombosis were at twice the risk of individuals with lower values compared to healthy controls [18]. In addition, research on individuals with a history or increased risk of cardiovascular disease has shown that elevated levels of FXI are linked to an increased chance of stroke, transient ischemic attack, myocardial infarction and serious coronary arterial disease, indicating that these levels raise the risk of thrombosis [19,20]. Inspired by this, inhibiting FXIa or FXI is expected to cause a lower risk of bleeding than warfarin or NOACs [13]. Coagulation factor XIa (FXIa) is emerging as a major target for anticoagulant treatment [21,22]. Factor XI (FXI), a serine protease homodimer, participates in the endogenous coagulation pathway. In reaction to vascular damage, factor XIIa or thrombin activates FXI to produce FXIa. Coagulation factor IXis primarily activated to factor IXa by FXIa, followed by the activation of factor X to FXa in the coagulation pathway of coagulation [6], which ultimately leads to the production of thrombin. Additionally, FXIa activates factors V, VIII, and X and promotes protein hydrolysis of the tissue factor pathway inhibitor (TFPI) to regulate thrombin production [23,24,25]. (Figure 1) By lowering the pressure of vascular and thromboembolic illness while preserving hemostasis, the targeted inhibition of FXIa provides a novel anticoagulant mechanism for the entire body with an enhanced benefit–risk profile [26]. Over the last decade, active site peptidomimetic inhibitors, polypeptides, antibodies, allosteric inhibitors, and aptamers have been found or developed as factor XI/XIa protein inhibitors. Additionally, antisense oligonucleotides that fundamentally lessen the hepatic production of factor XI have been developed [27]. Clinical trials of targeted anticoagulants against XIa have evolved considerably in the recent past. This review summarizes the development of anticoagulants and recent advances in clinical trials of experimental factor XI inhibitors from a clinical application perspective (Figure 1).

## 2. Results

As of 1 January 2023, our search screening included 33 clinical trials. Of these, 22 had been completed and had published results; 6 had been completed but the results had not been published, and 5 were enrolling (Table 1 and Table 2). There were seven clinical trials in the literature that evaluated the efficacy and safety of FXIa inhibitors [13,28,29,30,31,32,33].

### 2.1. ASO

Antisense oligonucleotides (ASOs) are represented by IONIS–FXIRX (BAY2306001), and FXI-LICA (BAY2976217). ASOs are relatively short lengths of unaltered or chemically altered nucleic acid sequences with single strand that can be designed to hybridize with target messenger RNA (mRNA) via a specific hydrogen combination or hydrophobic interactions to form heterodimers, which inhibit the synthesis of FXI and ultimately lowers its plasma concentration. ASOs are superior to traditional anticoagulants in that they require a lower dosing frequency and fewer medication interactions, have better target selectivity, and reduce drug design costs. Furthermore, serious bleeding is no more common than the large bleeding brought on by warfarin or LMWH medication [49].

#### IONIS-FXIRX (BAY2306001)

Research published in 2015 evaluated the efficiency and security of IONIS-FXIRX compared with Enoxaparin in 300 subjects having undergone total hip arthroplasty. They were randomly assigned 200 or 300 mg of FXIa inhibitor or 40 mg of enoxaparin. According to the findings, venous thrombosis occurred in 21 of 69 subjects (30%) who took Enoxaparin, 3 of 71 (4%) who took the 300 mg dose of IONIS-FXIRX and 36 of 134 (27%) who took the 200 mg dose. A recently released phase 2 trial of IONIS-FXIRX examined its safety and tolerability in subjects with end-stage renal disease needing hemodialysis (HD) (BAY2306001). Patients in the first of the two-part research received one open-label, 300 mg subcutaneous dose after hemodialysis followed by a second dose 28 days later to see how hemodialysis affected the medicine. For 12 weeks, participants in Part 2 were randomly assigned 200 or 300 mg or a placebo in a 1:1:1 allocation ratio. The research in Part 1 showed no significant distinction in the concentration-time profile of IONIS-FXIRX either after or before HD. When IONIS-FXIRX was administered before HD, T_max_ was 5.97 (range 2.98–10.0) and 6.03 (range 4.00–9.95), respectively. The second part of the trial revealed that mean FXI activity levels in the 300 and 200 mg and placebo groups at day 85 had declined compared to their baseline values by 56.0, 70.7, and 3.9%, respectively. FXI activity and FXI antigen levels were equal. The 200 and 300 mg doses brought about a considerable decrease in venous thrombosis due to the dialyzer compared to the placebo. Major bleeding events occurred less often in the 200 mg group (0%) than in the 300 mg (16.7%) or placebo group (17.7%) in patients taking IONIS-FXIRX during the randomization phase. There was no association with significant or clinically significant non-major bleeding incidents or any major adverse medication reactions [34].

### 2.2. Small Molecule Drugs

The small molecule drugs were represented by Milvexian (JNJ70033093, BMS-986177), Asundexian (BAY 2433334), BMS-986209, BMS-962212, ONO-7684 and SHR2285. Their mechanism of action is reversible binding to the catalytic domain of FXIa and blocking its function.

#### 2.2.1. BMS-962212

In 2018, published research examining the safety, pharmacodynamics, and pharmacokinetics of BMS-962212 in normal non-Japanese and Japanese subjects marked the beginning of clinical investigations into small-molecule FXIa inhibitors. In Part A, four doses of BMS-962212 or a placebo were administered for a 2 h intravenous infusion, and in Part B there were four doses of BMS-962212 or a placebo for a 5 day intravenous infusion on both non-Japanese (*n* = 4, *n* = 1 placebo) and Japanese (*n* = 4, *n* = 1 placebo) participants. The study’s findings demonstrated effective resistance to BMS-962212 with a quick pharmacodynamic (PD) response and quick elimination. FXI coagulation activity was reduced and at the same time exposure-dependent APTT rose. Japanese and non-Japanese subjects showed no discernible variations in pharmacokinetics or PD [35].

#### 2.2.2. Milvexian (JNJ70033093, BMS-986177)

Clinical studies of Milvexian have expanded in recent years. In one study, 1242 subjects who had undergone knee arthroplasty were randomly administered Milvexian (25, 50, 100 or 200 mg once or twice daily) or Enoxaparin (40 mg once a day). Milvexian inhibition of FXIa was discovered to be efficient in preventing venous thromboembolisms and reducing the rate of hemorrhaging. The occurrence rates of venous thrombogenesis in patients receiving 25, 50, 100 and 200 mg twice daily were 21, 11, 9 and 8%, respectively. In the once daily dose group, the occurrence rates for the 25, 50 and 200 mg doses were 25, 24 and 7%, respectively. In comparison, 54 of 252 patients who took Enoxaparin (21%) had venous thromboembolism, while 38 of 923 patients (4%) receiving Milvexian and 12 of 296 (4%) on Enoxaparin had some degree of hemorrhaging. Major hemorrhaging or clinically relevant non-major hemorrhaging occurred at rates of 1 and 2%, respectively, while severe adverse events occurred at rates of 2 and 4%, respectively [28]. Perera et al. found that 60 mg of Milvexian was typically well tolerated with no major adverse events (AEs), hemorrhagic adverse events (HAEs), or discontinuations owing to AEs in individuals with mild or moderate disorders or in the general population. For mild hepatic impairment in contrast to healthy liver function, the geometric mean ratios of maximum plasma concentration and area under the plasma concentration-time curve for Milvexian were 1.18 (95% CI: 0.735–1.895), 1.168 (95% CI: 0.725–1.882), 1.140 (0.699–1.857) and 0.996 (0.609–1.628). In contrast to participants with healthy liver function, there was no tendency towards long-term clearance in subjects with mild or moderate liver injury. Pharmacokinetic changes also suggested that patients having light or moderate liver injury were unlikely to require dose adjustment [36]. A similar study to the previous one by Perera et al. involving a single oral dose of 60 mg Milvexian was conducted in subjects with moderate (eGFR 30–59 mL/min/1.73 m^2^) or serious (eGFR < 30 mL/min/1.73 m^2^) renal impairment and in healthy populations. According to the findings, the maximum Milvexian concentration was comparable across all groups. In those with normal renal function, the area under the curve (AUC) values of Milvexian concentrations were 41 and 54% higher than the eGFR values of 30 and 15 mL/min/1.73 m^2^, respectively. In each group, the average time to achieve peak concentration was comparable. Compared with those with healthy renal function (13.8 h), the half-life was longer in subjects with moderate (18.0 h) or serious (17.7 h) renal disorders. This showed that Milvexian is safe and well tolerated in people with renal dysfunction [37]. A dose-exploration trial of Milvexian examined the tolerability, safety, PD and PK characteristics of sequential single incremental doses (SADs) and multiple incremental doses (MADs). According to the findings, Milvexian was typically secure and well tolerated at single and repeated incremental doses of up to 500 mg with no reports of clinically severe hemorrhaging. Milvexian was well tolerated by participants in good health after being administrated SADs of 4–500 mg (up to 200 mg twice a day) and MADs of 500 mg once a day. The PK profile of Milvexian was absorbed more rapidly (T_max_ 2–4 h) and the observed terminal elimination half-lives were 8.3–13.8 h in the 4–500 mg SAD group, 11.8 h in the 200 mg SAD group and 11.4–18.1 h in the MAD group, supporting twice daily or once daily dosing. With very little renal excretion and a dose of 20–200 mg, which is typically proportionate to the dose, the predominant route of elimination was through cytochrome P450 (CYP) metabolism, notably CYP3A. While Milvexian plasma concentrations declined with time, so did the severity of aPTT prolongation. The FXI coagulation activity showed an exposure-dependent decline in activity, which was consistent with the aPTT data [38]. According to Japanese research evaluating the safety, tolerability, PK, and PD of Milvexian using repeated incremental oral doses, 500 mg per day for 14 days was usually well tolerated with no fatalities, significant adverse events, or discontinuations as a result of adverse events. The median Tmax at the 50–200 mg dose group was comparable, and Milvexian exposure at 50–200 mg increased along with the dose (2.5–3.0 h). Repeated oral Milvexian doses led to decreased FXI coagulation activity and aPTT concentration-related prolongation. Its PK and PD profiles in Japanese patients were generally comparable to those previously seen in other populations, making them appropriate for continued clinical research of dosage profiles in Japanese [39]. Perera et al. examined the pharmacodynamic and pharmacokinetic effects of Diltiazem, Rifampin, and Itraconazole for clinical research into drug interactions with Milvexian [40,41]. After the co-administration of Itraconazole, the findings revealed a considerable rise in Milvexian exposure, while a modest increase was seen in the Diltiazem group. Single-dose co-administration of Rifampicin and Milvexian did not show any significant change in Milvexian exposure compared to Milvexian alone. Multiple doses of Rifampicin and Milvexian resulted in significantly lower-peak and total Milvexian exposure compared to a single dose of Milvexian. This is consistent with CYP3A metabolism and the involvement of P-glycoprotein in drug absorption and elimination. Mukul Sharma et al. are conducting a randomized, double-blind, placebo-controlled phase II trial to examine the dose response of Milvexian in individuals with transient ischemic attack (TIA) or ischemic stroke (NCT03766581). In participants undergoing dual antiplatelet treatment, the findings will demonstrate a dose–response relationship for Milvexian and provide a dosage basis for a larger phase III study of stroke prevention.

#### 2.2.3. SHR2285

SHR2285 is an oral FXIa inhibitor first developed in China by Jiangsu Hengrui Pharmaceutical Co., Ltd. (Lianyungang, China). According to Chen et al., a single oral dose of SHR2285 (50–400 mg) was well tolerated and safe in healthy volunteers, and no hemorrhagic events were seen. According to PK findings, SHR2285 had a mean half-life of 7.6–15.8 h and acted rapidly (median time to maximum plasma concentration varied from 3.0 to 4.0 h at various doses). SHR2285 has a short plasma half-life, which encourages further research into its potential as an acute anticoagulation treatment. In this clinical investigation, its PD parameters—prolonged aPTT, unchanged PT and INR after dosing—were comparable to those previously described for small-molecule FXIa inhibitors. This outcome was consistent with safety data showing no instances of hemorrhaging [43]. In addition, 52 healthy participants participated in a SHR2285 clinical study in China that also included aspirin, clopidogrel, or ticagrelor. Three treatment regimens were used to divide the participants into groups: aspirin and Clopidogrel plus placebo or SHR2285 200 mg, aspirin and Clopidogrel plus placebo or SHR2285 300 mg, and aspirin and Ticagrelor plus placebo or SHR2285 300 mg. For six consecutive days, twice daily oral doses were given to all groups. The average half-life of SHR2285 in groups A, B, and C were found to be 13.9, 14.5, and 13.8 h, respectively. In addition,, SHR2285 significantly inhibited FXI activity, with mean maximum inhibition rates of 84.8, 89.3, and 92.2 for FXI activity in the three groups, respectively. SHR2285 prolonged aPTT with good tolerance in normal individuals, and there was no proof that co-administration increased the rate of serious hemorrhaging [44].

#### 2.2.4. Asundexian (BAY 2433334)

A very important drug in the category of small-molecule inhibitors is Asundexian (BAY 2433334), an oral formulation that prolongs aPTT, inhibits FXIa, reduces arterial thrombus and is independent of antiplatelet agents [50]. Asundexian (BAY 2433334) was also used in the largest completed clinical trial of an FXIa inhibitor, the PACIFIC-AF study, which compared 20 and 50 mg of Asundexian with typical doses of Apixaban in 753 patients with atrial fibrillation to ascertain the best Asundexian dose. The results showed that in 251 subjects, Asundexian 20 mg generated 81% suppression of FXIa at trough concentrations and 90% reduction of FXIa activity at peak concentrations. At trough values, for Asundexian 50 mg, 254 subjects exhibited 92% inhibition and 94% inhibition at peak concentrations. Major hemorrhaging or clinically significant non-major hemorrhaging composite events were found for 3 subjects in the 20 mg group, 1 in the 50 mg group, and 6 in the Apixaban group according to the criteria of the International Society for Thrombosis and Haemostasia (ISTH). For these events, the overall Asundexian vs. Apixaban incidence ratio was 0.33 (90% CI: 0.09–0.97). Overall, bleeding rates were significantly lower and well tolerated in the Asundexian treatment group compared to Apixaban [29]. The aim of PACIFIC-Stroke was to evaluate the effectiveness and safety of Asundexian in preventing occult and symptomatic cerebral infarction in individuals with acute non-cardiogenic embolic ischemic stroke. It was a Phase 2b, randomly assigned, double-blind, placebo-controlled, dose-finding research. The main efficacy outcome was an MRI diagnosis of hidden cerebral infarction and recurrent symptomatic ischemic stroke at or before 26 weeks following allocation to the group. Of the 456 subjects in the placebo group, 87 (19%) experienced the main efficacy outcome, compared to 87 of the 455 participants in the Asundexian 10 mg group (19%) (crude incidence ratio: 0.99; 90% confidence interval: 0.79–1.24), 99 of the 450 participants in the Asundexian 20 mg group (22%) (incidence rate ratio 1.15; 90% CI: 0.93–1.43). In 90 of 477 subjects (20%) (incidence rate ratio 1.06; 90% CI: 0.85–1.32). 11 people in the placebo group (2%) had the main safety result, compared to 19 in the 10 mg group (4%), 14 in the 20 mg group (3%), and 19 in the 50 mg group (4%). For all Asundexian dosage groups, the pooled vs. placebo risk ratio was 1.57 (90% CI: 0.91–2.71), notwithstanding the fact that neither the safety result nor the main effectiveness outcome showed a discernible distinction between the Asundexian and placebo groups. However, the results of the post hoc analysis suggested that suppression of FXIa with Asundexian decreased reoccurring ischemic stroke and transient ischemic attacks without increasing bleeding in patients with acute non-cardiogenic embolic ischemic stroke compared with placebo, especially in individuals with atherosclerosis [30]. In a placebo-controlled, phase 2, double-blind research by Sunil V Rao et al. [31], 1610 patients receiving dual antiplatelet therapy after recent acute myocardial infarction (MI) were randomly assigned a 10, 20 or 50 mg oral dose of Asundexian or a placebo once a day for six months to a year. At 4 weeks, the effect of Asundexian on FXIa inhibition was evaluated. Primary safety and efficacy outcomes were assessed at follow-up. Asundexian produced dose-related suppression of FXIa activity: 50 mg produced a reduction of >90%, according to the results. When given a 10, 20, 50 mg dose or a placebo, the primary safety outcome occurred in 30 (7.6%), 32 (8.1%), 42 (10.5%) and 36 (9.0%) of individuals, respectively. The combined Asundexian-to-placebo risk ratio was 0.98 (90% CI: 0.71–1.35). Patients randomly given Asundexian 10, 20, 50 mg or a placebo, respectively, had the following efficacy outcomes: 27 (6.8%), 24 (6.0%), 22 (5.5%), and 22 (5.5%). These findings confirmed the safety and effectiveness of Asundexian after an MI.

### 2.3. Antibody

The representative antibodies Abelacimab (MAA868), Osocimab (BAY1213790), and AB023 (Xisomab) bind to the catalytic site of Factor XI and lock it in the zymogen (inactive precursor) form, preventing Factor XIIa or thrombin from activating it, which has the effect of inhibiting Factor XI.

#### 2.3.1. Abelacimab (MAA868)

Patients having undergone total knee arthroplasty participated in an open-label parallel trial to evaluate the effectiveness and safety of Abelacimab (30, 75 or 150 mg) vs. Enoxaparin (40 mg). Venous thromboembolism was found in 13 of 102 patients (13%) in the 30 mg Abelacimab group, 5 of 99 (5%) in the 75 mg group, 4 of 98 (4%) in the 150 mg group, and 22 of 101 (22%) in the Enoxaparin group. Moreover, there was little major hemorrhaging or clinically relevant non-major hemorrhaging in any of the experimental groups. At the higher dose of Abelacimab compared to Enoxaparin, thrombosis incidence and severity decreased [32]. Both Abelacimab and antisense oligonucleotides were superior to Enoxaparin in preventing a postoperative venous thromboembolism; however, antisense oligonucleotides must be used for one month prior to surgery to decrease FXI to therapeutic concentrations, whereas the functional factor XI level decreased within minutes following intravenous administration of Abelacimab, which permitted its postoperative administration. Byungdoo Alexander Yiet al. explored the pharmacodynamics (PD), pharmacokinetics (PK) and safety of single intravenous and multiple subcutaneous injections of Abelacimab in healthy individuals and patients with atrial fibrillation (AF). Results from studies in ANT-003 (single intravenous injections of 30, 50 and 150 mg Abelacimab in healthy subjects) and ANT-004 (subcutaneous injections of 120 and 180 mg Abelacimab once a month in patients with AF) showed Abelacimab to have good safety and tolerance, with no severe adverse reactions or deaths, and it generated a substantial drop in free factor XI from initial assessment, which persisted throughout the monthly dosage period [45]. Two ongoing phase 3, multicenter, randomized, open-label blinded clinical trials are comparing the effects of Abelacimab with Dalteparin on reappearance and hemorrhaging in subjects with genitourinary-related and gastrointestinal venous thrombosis (NCT05171075), and Abelacimab with Apixaban on recurrence and bleeding in cancer-related venous thromboembolisms.

#### 2.3.2. Osocimab (BAY1213790)

Osocimab is a monoclonal G1 antibody that has been completely humanized and is intended to target and neutralize FXIa. According to crystal structure research, its metastable conformation has a new mode of action, which shows that binding to areas close to the FXIa active site results in significant structural reorganizations [51]. Osocimab was discovered to demonstrate a dose-dependent increase in aPTT and a dose-dependent reduction in FXI activity after intravenous treatment in a phase 1 research by Thomas et al. [46]. Up to six days following trial administration, bleeding times were measured and neither the Osocimab group nor the placebo group experienced any longer bleeding periods. The average time to reach the highest plasma concentration was shown to be one to four hours. It was discovered that plasma drug concentrations and concentration-time profiles rose with the dosage. The volume of distribution and overall low clearance were in line with other monoclonal antibody pharmacokinetic profiles. The average half-life of elimination was 30–44 days. The researchers investigated several dosages of Osocimab in comparison to Apixaban and Enoxaparin in patients having undergone knee arthroplasty in a randomized, transparent, blinded, non-inferiority study to determine the optimal balance of effectiveness and safety. The primary endpoints symptomatic DVT, asymptomatic DVT by required bilateral venography, and non-fatal pulmonary embolism were proven. The primary safety outcome was the occurrence rate of clinically relevant hemorrhaging. The primary efficacy outcomes of the FOXTROT trial were as follows: In the Osocimab group, 14 patients (17.9%) taking 1.8 mg/kg, 13 patients (16.5%) taking 1.2 mg/kg, 8 patients (15.7%) taking 0.6 mg/kg, and 18 patients (23.7%) taking 0.3 mg/kg. In the preoperative Osoximab group, the occurrence frequency of the primary outcome was 23 patients receiving 0.3 mg/kg (29.9%) and 9 patients receiving 1.8 mg/kg (11.3%). In contrast, 20 patients receiving Enoxaparin (26.3%) and 12 patients receiving Apixaban (14.5%) were observed for the primary outcome. Postoperative severe or clinically relevant non-major hemorrhaging 10–13 days after operation occurred in 3% of the 1.8 mg/kg group, 1% of the 1.2 mg/kg group and 2% of the 0.3 mg/kg group; similarly, hemorrhaging occurred in 1.9% of the 0.3 mg/kg and 4.7% of the 1.8 mg/kg preoperative groups. In contrast, 5.9% of Enoxaparin patients and 2% of Apixaban patients experienced primary hemorrhagic events compared to none in the 0.6 mg/kg Osocimab group [33]. For individuals with end-stage renal disease (ESRD) taking routine dialysis, a double-blind, multi-center, randomly assigned, placebo-controlled, parallel clinical trial is being conducted to evaluate the safety of Osocimab at low (105 mg) and high (210 mg) doses (NCT04523220). In addition, investigators in another observer-blind, multi-center, placebo-controlled parallel clinical trial are assessing the safety and tolerability of Osocimab in individuals with ESRD undergoing hemodialysis (NCT03787368).

#### 2.3.3. AB023 (Xisomab)

AB023 (Xisomab) is a recombinant humanized antibody that combines to the apple2. These other pathways of FXI activation are unaffected by AB023, and FXIa activation of FXI is not prevented. Thus, AB023 works as an FXIIa inhibitor despite binding to FXI [52,53,54]. To access the pharmacology and safety of AB023, Lorentz et al. [47] carried out a phase 1 placebo-controlled, randomized, double-blind clinical trial. In this single incremental dosage research, AB023 was administered intravenously to 16 healthy individuals at doses ranging from 0.1 to 5 mg/kg, while 5 other participants took a placebo. The severity and frequency of adverse events that developed as a result of the therapy were the primary endpoints. The pharmacokinetics, pharmacodynamics, and immunogenicity of the medication were some of the ancillary objectives. AB023 was generally well tolerated and caused no severe negative effects. Compared to 3 of 5 patients receiving placebo(60%), 7 of 16 patients taking AB023 (44%) experienced acute adverse events. None of the patients who received AB023 produced anti-drug antibodies. This trial, however, was too small to determine infrequent but possibly significant side effects. A dose of 0.1 mg/kg of free AB023 had a half-life of about 1.3 h according to the pharmacokinetic data, whereas a dose of 5 mg/kg had a half-life of about 121 h. The outcomes also showed that AB023 extended aPTT in a dose-dependent manner. The lowest dosage of AB023 needed to cause an almost two-fold aPTT extension was 0.5 mg/kg [47]. A randomly-assigned, double-blind, phase 2 study compared the safety and effectiveness of AB023 with placebo in 24 subjects with ESRD taking heparin-free hemodialysis. The frequency and seriousness of adverse events were the primary safety result. Visual scores of clots in the dialyzer and venous cavity were the primary efficacy result. The results showed no clinically relevant hemorrhagic events, but viral gastroenteritis, abdominal pain and diarrhea occurred. The mean pre-dose incidence of high-level dialyzer clots was 83, 75 and 87.5% in the placebo, 0.25 mg/kg AB023 and 0.5 mg/kg groups, respectively. Prothrombin-antithrombin complex and C-reactive protein levels were relatively low after AB023 treatment compared to the data gathered before administration. There was also a decreased frequency of occlusive episodes necessitating hemodialysis circuit replacement. Along with reducing blood accumulation in the dialyzer, AB023 significantly decreased potassium and iron retention [48]. The effectiveness of AB023 in avoiding catheter-associated thrombus formation in cancer patients undergoing chemotherapy is being studied in a phase II study. The outcomes will offer crucial proof in support of using AB023 in chemotherapy patients to reduce catheter-induced thrombosis (NCT04465760).

### 2.4. Safety Meta-Analysis of XIa Inhibitors

#### 2.4.1. Efficacy

The primary efficacy outcomes were the exploratory thrombotic composite endpoints of the occurrence of systemic venous thromboembolism, symptomatic ischemic stroke, occult cerebral infarction and myocardial infarction. The results showed no statistically meaningful distinction in the primary efficacy outcome between patients receiving FXIa inhibitors compared to controls in seven studies. Figure 2 shows a forest map of the main therapeutic outcomes (RR = 0.796; 95% CI: 0.606–1.046; I^2^ = 68%), and Figure 3 shows a funnel chart. According to Egger’s test t = −0.59; *p* = 0.5769 (*p* > 0.1), there was no publication bias.

#### 2.4.2. Risk of Bleeding

We combined the bleeding risk outcomes for hemorrhagic events, serious hemorrhaging and clinically relevant hemorrhaging into separate effect sizes. The outcomes did not indicate a statistical difference in bleeding in patients receiving FXIa inhibitors and controls in seven studies (Figure 4: RR = 0.717; 95% CI: 0.502–1.023; I^2^ = 60%). Egger’s test (t = −0.81, *p* = 0.4486 (*p* > 0.1) showed no publication bias (Figure 5). There was no statistically significant distinction in the frequency of substantial bleeding and clinically relevant hemorrhaging in patients receiving FXIa inhibitors compared to control groups (Figure 6: RR = 0.709; 95% CI: 0.430–1.169; I^2^ = 50%). Egger’s test resulted in t = −0.98; *p* = 0.3638 (*p* > 0.1), and there was no publication bias (Figure 7). Subgroup analysis found significant distinctions in severe bleeding and clinically relevant hemorrhaging in subjects receiving FXIa inhibitors in contrast to Enoxaparin (Figure 8: RR = 0.457; 95% CI: 0.256–0.816; I^2^ = 0%).

## 3. Methods

### 3.1. Search Strategy

To identify relevant studies, we searched Embase, PubMed, and Cochrane Library as well as the Clinical Trial Database. The search was based on a fusion of subject and free terms and was adapted to the characteristics of each database. References included in the study were also searched to supplement access to relevant material. We then also conducted a manual search based on the clinical trial registry. Our search in PubMed was as follows: (((((((((Factor XIa Inhibitors) OR (Factor Eleven A Inhibitors)) OR (Activated Factor XI Inhibitors)) OR (Coagulation Factor Xia Inhibitors)) OR (Factor XIa, Coagulation Inhibitors)) OR (Blood Coagulation Factor XI, Activated Inhibitors)) OR (Contact Activation Product Inhibitors)) OR (Factor 11A Inhibitors)) OR (Factor XI, Activated Inhibitors)) AND (clinical trial).

### 3.2. Literature Screening

By reading the title abstracts we excluded non-human research, case reports, literature reviews and irrelevant publication types. Additionally, we did not include any non-English language publications. After screening for inclusion by reading the full texts, we focused on the clinical trials of factor XI inhibitors, both completed and incomplete. We summarized the included literature, including author, year, drug type, drug name, participating population, and clinical registry number.

### 3.3. Statistical Analysis

We performed a quantitative analysis of the safety and efficacy of FIXa factor inhibitors by meta-analysis. The Mantel–Haenszel with a random effects model was used to evaluate pooled effect sizes for the meta-analyses. Relative risk ratios (RRs) were reported with 95% confidence intervals (CIs). To measure between-study heterogeneity, the Cochran’s test and I^2^ tests were used. At *p*  <  0.10 and I^2 ^ >  50%, statistically significant heterogeneity was determined to be present [55,56]. For each result, forest plots were made. Visual evaluation of publication bias was done using funnel plots and the Egger test. All analyses were carried out using R 4.1.1. The statistical analysis considered a *p* < 0.1 to be significant.

## 4. Summary

It is believed that medications that target FXIa may induce efficient thromboprophylaxis without raising the risk of bleeding if intrinsic pathways, such as those leading to the synthesis of thrombin and fibrin, are effectively inhibited. The effectiveness of FXI inhibition has received adequate proof of concept from clinical studies. Peptides, active-site peptidomimetic and metabotropic inhibitors, ASOs, aptamers, and monoclonal antibodies are all methods for blocking the FXI/FXIa system to provide a range of treatment alternatives to address the requirements of thrombosis patients. For instance, whereas ASO, aptamers, and antibodies are appropriate for parenteral delivery, active-site and variant FXIa inhibitors have the potential for oral or parenteral usage. In addition, the relatively delayed and long duration of action of ASOs were expected, which makes them suitable for chronic observation.

In the quantitative analysis section, the small number of available clinical trials meant that the FXIa inhibitor group was only crudely pooled with placebo or other drug controls instead of being linked precisely to a particular dose of a particular drug for FXI inhibitors. However, the results all suggested that FXIa inhibitors are as effective and safe as earlier anticoagulants. Nevertheless, with direct inhibition of FXIa and thrombin, FXI inhibition also includes an off-target effect that might be associated with vasoprotective properties due to protease-activated receptor (PAR) cell signaling modifications as well as other complex diseases such as diabetes, nephropathy, fibrosis and cancer, which may be regulated by PAR [57]. In addition, there may be other specific effects of FXIa that need to be considered. In addition to the primary substrate FXI, other coagulation factors including FV, FVIII, and FX can be activated by FXIa. FXIa proteins have the ability to cleave tissue factor pathway inhibitors (TFPI) hydrolytically, which may boost the tissue factor pathway’s ability to generate thrombin [58,59]. 

Despite encouraging developments in several FXI/FXIa-targeted therapy areas, substantial work is still required to deliver these treatments to the clinic. Most clinical trials that target FXI/FXIa seem to be centered on selecting the best administration method, medication dose, drug interaction and therapeutic effectiveness and safety assessments. In the future, a potent antidote could also be required to deal with cases of excessive anticoagulation. The findings of the clinical studies using FXI/FXIa inhibitors will substantially promote their broad clinical use given the tremendous focus this protein target has received over the past 10 years.

## Figures and Tables

**Figure 1 pharmaceuticals-16-00866-f001:**
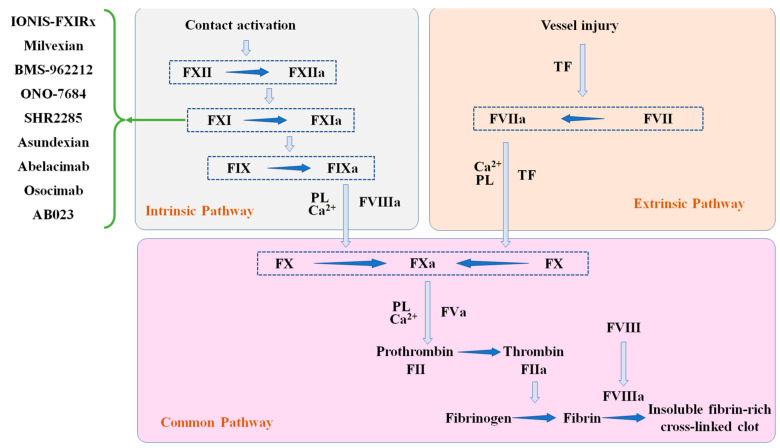
Endogenous and exogenous anticoagulation pathways.

**Figure 2 pharmaceuticals-16-00866-f002:**
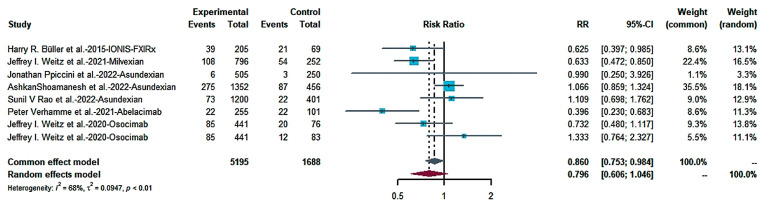
Forest map of the main therapeutic outcomes [13,28,29,30,31,32,33].

**Figure 3 pharmaceuticals-16-00866-f003:**
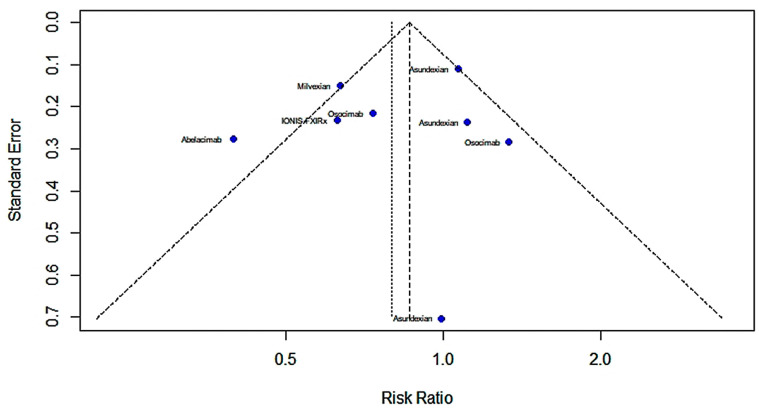
Funnel chart of main therapeutic outcomes [13,28,29,30,31,32,33].

**Figure 4 pharmaceuticals-16-00866-f004:**
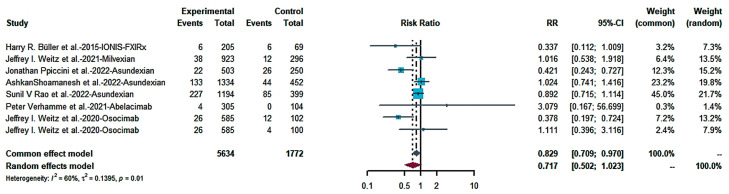
Forest map of any bleeding [13,28,29,30,31,32,33].

**Figure 5 pharmaceuticals-16-00866-f005:**
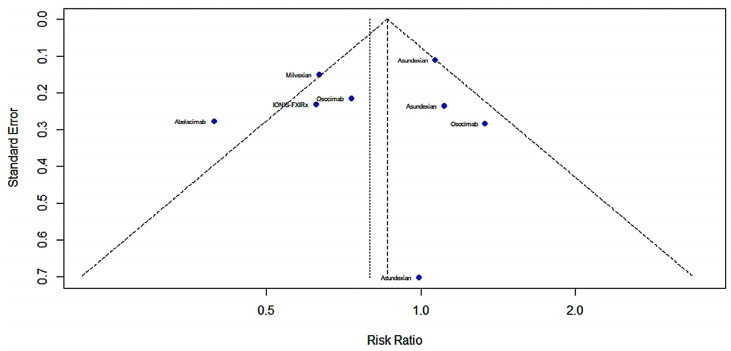
Funnel chart of any bleeding [13,28,29,30,31,32,33].

**Figure 6 pharmaceuticals-16-00866-f006:**
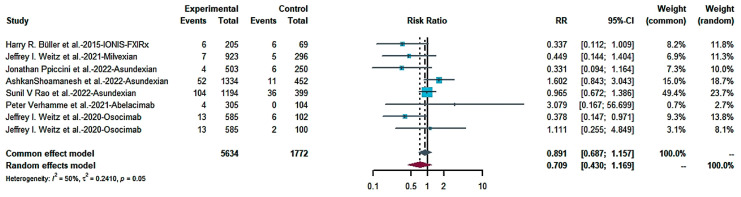
Forest map of major hemorrhage and clinically relevant hemorrhage [13,28,29,30,31,32,33].

**Figure 7 pharmaceuticals-16-00866-f007:**
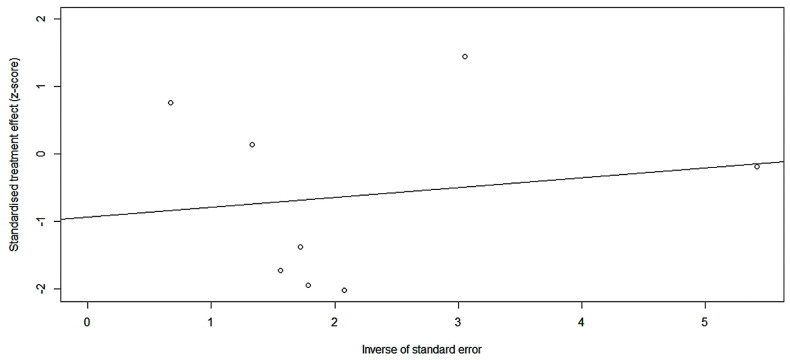
Egger chart of major hemorrhage and clinically relevant hemorrhage [13,28,29,30,31,32,33].

**Figure 8 pharmaceuticals-16-00866-f008:**
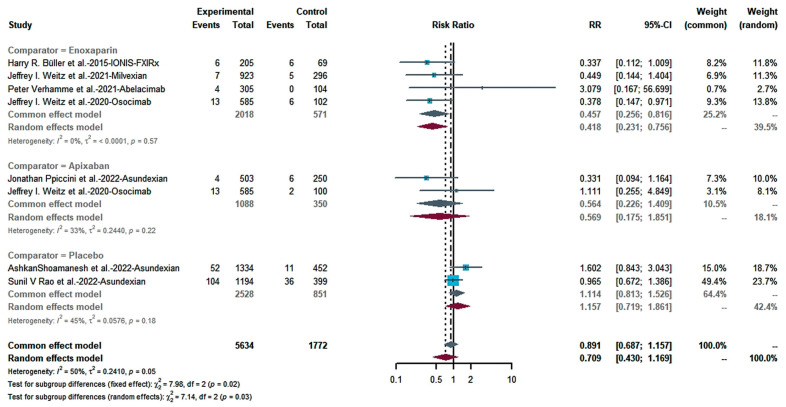
Subgroup analysis forest map of major hemorrhaging and clinically relevant hemorrhaging [13,28,29,30,31,32,33].

**Table 1 pharmaceuticals-16-00866-t001:** Completed and published clinical studies.

Number	Author	Year	Compound	Type	Comparator	Subject	Age	Total Number	Study Design	Trial Registration	Status
1 [34]	Michael Walsh et al.	2022	IONIS-FXIRx	ASO	placebo	end-stage renal disease	29–80	49	Part1: open label, phase 2 Part2: randomized, double-blin, phase 2	NCT02553889	published
2 [13]	Harry R. Büller et al.	2015	IONIS-FXIRx	ASO	Enoxaparin	undergoing total knee arthroplasty	18–80	293	randomized, open-label, parallel group, phase 2	NCT01713361	published
3 [35]	Vidya Perera et al.	2018	BMS-962212	small molecular	placebo	Japanese and non-Japanese healthy subjects	18–45	74	randomized, double-blind, sequential, ascending dose, placebo-controlled, phase 1	NCT03197779	published
4 [28]	Jeffrey I. Weitz et al.	2021	Milvexian	small molecular	Enoxaparin	undergoing total knee arthroplasty	50–90	1242	randomized, parallel-group, open label, phase 2	NCT03891524	published
5 [36]	Vidya Perera et al.	2022	Milvexian	small molecular	none	patients with liver injury healthy subjects	18–70	26	nonrandomized, single oral dose, parallel-group, open-label, phase 1	NCT02982707	published
6 [37]	Vidya Perera et al.	2022	Milvexian	small molecular	none	moderate or severe renal impairment	18–70	43	nonrandomized, open-label, parallel-group, single dose, phase 1	NCT03196206	published
7 [38]	Vidya Perera et al.	2022	Milvexian	small molecular	placebo	healthy subjects	18–55	104	randomized, double-blind, placebo-controlled, sequential, SAD and MAD, phase 1	NCT02608970	published
8 [39]	Vidya Perera et al.	2022	Milvexian	small molecular	placebo	healthy subjects in Japanese	18–55	33	randomized, double-blind, placebo-controlled, multiple ascending-dose, phase 1	NCT03224260	published
9 [40]	Vidya Perera et al.	2022	Milvexian	small molecular	none	healthy subjects	18–55	28	nonrandomized, open label, two-period, crossover, phase 1	NCT02807909	published
10 [41]	Vidya Perera et al.	2022	Milvexian	small molecular	none	healthy subjects	18–55	16	nonrandomized, open-label, single-sequence, phase 1	NCT02959060	published
11 [42]	Dominic Beale et al.	2021	ONO-7684	small molecule	placebo	healthy subjects	18–55	72	randomized, double-blind, single and multiple dose, phase 1	NCT03919890	published
12 [43]	Rui Chen et al.	2022	SHR2285	small molecular	placebo	healthy subjects	15–45	28	randomized, double-blind, dose-ascending, single-dosing, phase 1	NCT03769831	published
13 [44]	Tingting Ma et al.	2022	SHR2285	small molecular	placebo	healthy subjects	18–55	52	randomized, double-blind, single-center, placebo-controlled, phase 1	NCT04945616	published
14 [29]	Jonathan Ppiccini et al.	2022	Asundexian	small molecular	Apixaban	atrial fibrillation	mean: 75	753	randomized, double-blind, multicenter, phase 2	NCT04218266	published
15 [30]	AshkanShoamanesh et al.	2022	Asundexian	small molecular	placebo	non-cardioembolic ischaemic stroke	mean: 67	1808	randomized, double-blind, parallel-group, placebo-controlled, phase 2b	NCT04304508	published
16 [31]	Sunil V Rao et al.	2022	Asundexian	small molecular	placebo	acute myocardial infarction	mean: 65	1601	randomized, double-blind, parallel-group, phase 2	NCT04304534	published
17 [32]	Peter Verhamme et al.	2021	Abelacimab	antibody	Enoxaparin	undergoing total knee arthroplasty	18–80	412	randomized, parallel-group, prospective, phase 2	EudraCT 2019-003756-37	published
18 [45]	B. Alexander Yi et al.	2022	Abelacimab	antibody	placebo	healthy subjects and atrial fibrillation	ANT-003: 18–60 ANT-004: 18–85	ANT-003: 32 ANT-004: 18	randomized, subject/patient-and investigator-blinded, placebo-controlled, multiple ascending dose	none	published
19 [46]	Dirk Thomas et al.	2019	Osocimab	antibody	placebo	healthy subjects	18–55	83	randomized, single-blind, parallel-group, placebo-controlled, dose-escalation, phase 1	EudraCT 2014-003816-35	published
20 [33]	Jeffrey I. Weitz et al.	2020	Osocimab	antibody	Enoxaparin, Apixaban	undergoing total knee arthroplasty	mean: 66.5	813	randomized, open-label, parallel-group, phase 2	NCT03276143	published
21 [47]	Christina U Lorentz et al.	2019	AB023	antibody	placebo	healthy subjects	18–48	21	randomized, double-blind, placebo-controlled, single ascending bolus dose, phase 1	NCT03097341	published
22 [48]	Christina U Lorentz et al.	2021	AB023	antibody	placebo	end-stage renal disease (ESRD)	18–80	24	randomized, double-blind, placebo-controlled, single-dose, phase 2	NCT03612856	published

**Table 2 pharmaceuticals-16-00866-t002:** Unpublished clinical studies.

Number	Compound	Type	Comparator	Subject	Age	Total Number	Study Design	Trial Registration	Status
1	IONIS-FXIRX	ASO	placebo	end-stage renal disease (ESRD)	18–85	213	randomized, double-blind, placebo-controlled, phase 2	NCT03358030	Completed unpublished
2	FXI-LICA	ASO	placebo	end-stage renal disease (ESRD)	≥18	307	randomized, double-blind, placebo-controlled, phase 2	NCT04534114	Completed unpublished
3	Milvexian	small molecular	Placebo, Clopidogrel, Aspirin	Acute Ischemic Stroke, Transient Ischemic Attack (TIA)	≥40	2366	randomized, double-blind, placebo-controlled, dose-ranging, phase 2	NCT03766581	Completed unpublished
4	SHR2285	small molecular	Enoxaparin	undergoing elective unilateral total knee arthroplasty	40–75	500	randomized, double-blind, open-label, multicenter, positive-controlled, phase 2	NCT05203705	Recruiting
5	BMS-986209	small molecular	Placebo, Itraconazole, Diltiazem	healthy subjects	18–55	114	randomized, single and multiple doses, phase 1	NCT04154800	Completed unpublished
6	Abelacimab	antibody	Dalteparin	Gastrointestinal/Genitourinary Cancer and Associated VTE	≥18	1020	randomized, blinded endpoint evaluation, multicenter, phase 3	NCT05171075	Recruiting
7	Abelacimab	antibody	Apixaban	Cancer associated thrombosis	≥18	1655	randomized, blinded endpoint evaluation, multicenter, phase 3	NCT05171049	Recruiting
8	Osocimab	antibody	placebo	end-stage renal disease (ESRD)	≥20	686	randomized, double-blind, placebo-controlled, multicenter, parallel-group, phase 2	NCT04523220	Completed unpublished
9	Osocimab	antibody	placebo	end-stage renal disease (ESRD)	18–80	55	randomized, observer-blind, multicenter, placebo-controlled, parallel-group, phase 1	NCT03787368	Completed unpublished
10	AB023	antibody	none	patients with cancer receiving chemotherapy	≥18	50	single-group, open-label, phase 2	NCT04465760	Recruiting
11	MK-2060	antibody	placebo	end-stage renal disease (ESRD)	≥18	489	randomized, double-blind, placebo-controlled, multicenter, parallel-group, phase 2	NCT05027074	Recruiting

## Data Availability

The datasets used in this study are available from the corresponding author upon reasonable request.

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
