# Peer review of "Factor XIa Inhibitors as a Novel Anticoagulation Target: Recent Clinical Research Advances"

_pharmaceuticals, 2023, doi:10.3390/ph16060866_

Round 1

Reviewer 1 Report

The present article offers an overview of the development of anticoagulants, along with recent advances in clinical trials on existing factor XI inhibitors, from a clinical application perspective. The authors conducted an electronic search and performed a meta-analysis of early-phase trials. The manuscript aims to provide a comprehensive summary of the current state of research on anticoagulation therapy, including unmet needs and recent findings from clinical trials. The manuscript requires considerable editing before publication, including limiting the text length to 3500-5000 words and presenting the information in edited segments.

The manuscript should summarize the unmet needs that persist in the era of direct anticoagulants by citing data from clinical trials and summaries. Specifically, these include residual thromboembolic risk during DOAC anticoagulation, along with its various interactions with both platelet-related and unrelated mechanisms (as observed in myocardial infarction; see PMID: 31533437), high risk of bleeding events, particularly mucosal/GI bleeding, and the risks associated with concomitant use of antiplatelets (PMID: 36568540, PMID: 32418529). Additionally, details from aXI clinical trials should be limited to those relevant and related to the analyses.

The use of meta-analyses and statistical tests adds rigor to the findings. However, the article could benefit from more detailed descriptions of the methods used to evaluate the included literature. The manuscript mentions seven trials in the abstract, while the analyses show eight trials, but no numerical data is given in the abstract. Additionally, while the conclusion is supported by the results of the included studies, it would have been helpful to discuss important limitations of early-phase studies. Nevertheless, the article has the potential to provide valuable insights into the development of FXIa inhibitors in anticoagulation therapy.

Minor issues include the need to revise the vague statement about "clinically used anticoagulant drugs that inhibit the coagulation pathway directly or indirectly" and to use the term "experimental" instead of "existing" when referring to factor XI inhibitors. Also, the manuscript should use "identify" rather than "locate" when describing the search for relevant studies. The manuscript should specify that the Mantel-Haenszel methods are the default fixed-effect methods of meta-analysis and the caption of Figure 1 should be more informative.

Word usage in medical technical terms are frequently inadequate. Language editing with native speaker medical professional is warranted. 

Author Response

Comment 1: The manuscript should summarize the unmet needs that persist in the era of direct anticoagulants by citing data from clinical trials and summaries. Specifically, these include residual thromboembolic risk during DOAC anticoagulation, along with its various interactions with both platelet-related and unrelated mechanisms (as observed in myocardial infarction; see PMID: 31533437IF: 3.299 Q3 ), high risk of bleeding events, particularly mucosal/GI bleeding, and the risks associated with concomitant use of antiplatelets (PMID: 36568540IF: 5.846 Q2 , PMID: 32418529).

Response: Thank you very much for your suggestions on our manuscript. We have carefully read the literature you provided and added it to the article. We are confident that these additions to the literature will enrich our article.

Comment 2: The use of meta-analyses and statistical tests adds rigor to the findings. However, the article could benefit from more detailed descriptions of the methods used to evaluate the included literature. The manuscript mentions seven trials in the abstract, while the analyses show eight trials, but no numerical data is given in the abstract.

Response: Thank you very much for your suggestion. Seven studies were included in the meta-analysis section of this paper, but one of them compared Osocimab with both Enoxaparin and Apixaban. We treated this study as two trials for the meta-analysis.

Comment 3: Additionally, while the conclusion is supported by the results of the included studies, it would have been helpful to discuss important limitations of early-phase studies. Nevertheless, the article has the potential to provide valuable insights into the development of FXIa inhibitors in anticoagulation therapy.

Response: Thank you very much for your suggestions. Based on your suggestions, we have made changes and additions to the final summary section of the article.

Comment 4: Minor issues include the need to revise the vague statement about "clinically used anticoagulant drugs that inhibit the coagulation pathway directly or indirectly" and to use the term "experimental" instead of "existing" when referring to factor XI inhibitors. Also, the manuscript should use "identify" rather than "locate" when describing the search for relevant studies. The manuscript should specify that the Mantel-Haenszel methods are the default fixed-effect methods of meta-analysis and the caption of Figure 1 should be more informative.

Response: Thank you very much for your attention to the details of our manuscript, which we have revised one by one based on your suggestions. Thank you again for your conscientiousness and rigor.

Comment 5: The manuscript requires considerable editing before publication, including limiting the text length to 3500-5000 words and presenting the information in edited segments.

Response: Thank you very much for your suggestion, we have trimmed and modified the article, and the text is controlled to about 5500.

Reviewer 2 Report

The article entitled “Factor XIa inhibitors as a novel anticoagulation target: recent clinical research advances” is a review about the clinical trials of new anticoagulant molecules such as the FXI inhibitors. The authors report the informations in terms of route of administration, medication dose, drug interactions, assessment of therapeutic effectiveness and safety. This review is difficult to read. Therefore, I suggest to make some adjustments to the structure of this article. First, I would shorten the paragraph entitled “Background” and the paragraph entitled “The theoretical basis of anticoagulation involving FXI/FXIa”. Second, I would insert the “highlights” in the paragraphs entitled “Clinical studies of factor XIa inhibitors”, “Small molecular”, and “Antibody”. I think that the splitting up the paragraphs makes reading easier and it captures the reader’s attention. I think that these structural changes are necessary. Therefore, this article is not suitable for publication in its current version.

The English used is correct and readable.

Author Response

Comment 1: Therefore, I suggest to make some adjustments to the structure of this article. First, I would shorten the paragraph entitled “Background” and the paragraph entitled “The theoretical basis of anticoagulation involving FXI/FXIa”. Second, I would insert the “highlights” in the paragraphs entitled “Clinical studies of factor XIa inhibitors”, “Small molecular”, and “Antibody”. I think that the splitting up the paragraphs makes reading easier and it captures the reader’s attention. I think that these structural changes are necessary. Therefore, this article is not suitable for publication in its current version.

Response: Thank you very much for your suggestion, we have cut down the article and revised the paragraph structure according to your suggestion. We hope that these changes will make the article clearer and easier to read.

Reviewer 3 Report

In this meta-analysis, the authors addressed an important social problem which is represented by the poor efficacy or controindications of the common anticoagulants used today. The research work has been carried out with diligence and the results are original and clinically rilevant.

The study is more exhaustive than others published recently having evaluated a greater number of anticoagulant drugs with the principle of FXI inhibitors.

The conclusions are compatible with the results obtained, even if,a sufficient number of studies and patients were not considered for all the drugs evaluated.The antithrombotic effect and the reduction of haemorrhagic events found in patients treated with XI inhibitors is an important result which, however, needs, as confirmation, further investigations in different populations

The “small molecule” paragraph is complex to read and not very linear, a summary table  could be of help.

Author Response

Comment 1: The conclusions are compatible with the results obtained, even if,a sufficient number of studies and patients were not considered for all the drugs evaluated. The antithrombotic effect and the reduction of haemorrhagic events found in patients treated with XI inhibitors is an important result which, however, needs, as confirmation, further investigations in different populations.

Response: Thank you very much for the offer. It is a great idea and provides us with a good idea for future studies. In future studies, we will continue to focus on the safety and efficacy of the drug in different populations.

Comment 2: The “small molecule” paragraph is complex to read and not very linear, a summary table could be of help.

Response: Thank you very much for your suggestion, we have broken down the paragraph structure of this section. We hope that these changes will make the article clearer and easier to read.

Reviewer 4 Report

In this review, Xia et al summarize the development of anticoagulants and recent advances in clinical trials of existing FXI inhibitors. The review is interesting to read, but needs to be revised to organize it better: 

1. Methods: search strategy and literature screening sections are extraneous, as this is standard with all review articles. 

2. Section 2.4. Theoretical basis of anticoagulation involving FXI/FXIa: This doesn't seem to fit into the methods. Consider incorporating into the introduction? The introduction is very lengthy as well and needs to be condensed to convey a few most important main points. 

3. Figure 1: it may be helpful to dive deeper into mechanisms of each drug. Which compound inhibits FXI autoactivation, FXI activation by FXII, FXI activation by thrombin feedback loop. The current Figure 1 doesn't add much to current knowledge of the coagulation cascade. 

4. Table 1: It may be helpful to add in text citations for the published clinical studies

5. Please divide long sections into separate paragraphs with subheadings to organize the paper better. It is currently very difficult to read and be engaged. 

6. Many of the figures, especially forest maps, are so small/pixelated and impossible to read. 

Please check spelling/spacing/grammar throughout.

Author Response

Comment 1: Methods: search strategy and literature screening sections are extraneous, as this is standard with all review articles.

Response: Thank you very much for your suggestion, we feel that adding the search strategy and literature screening makes our article more rigorous and structured. Your suggestion is correct, but we would like to be more detailed and organized.

Comment 2: Section 2.4. Theoretical basis of anticoagulation involving FXI/FXIa: This doesn't seem to fit into the methods. Consider incorporating into the introduction? The introduction is very lengthy as well and needs to be condensed to convey a few most important main points.

Response: Thank you very much for your suggestion, and we agree with your proposal. We have streamlined and modified this section.

Comment 3:  Figure 1: it may be helpful to dive deeper into mechanisms of each drug. Which compound inhibits FXI autoactivation, FXI activation by FXII, FXI activation by thrombin feedback loop. The current Figure 1 doesn't add much to current knowledge of the coagulation cascade.

Response: Thank you very much for your suggestion, we have described the mechanism of inhibition of factor XI in the paragraphs for each type of drug separately. These mechanisms involve many molecules and are difficult to express accurately in detail in Figure 1. Thank you very much for your suggestion. We believe that in future studies, we can discuss in depth the detailed mechanisms behind these drugs and assess the differences between the different types of drugs.

Comment 4: Table 1: It may be helpful to add in text citations for the published clinical studies.

Response: Thank you very much for your suggestion, we have modified the table and added a citation inside the table.

Comment 5: Please divide long sections into separate paragraphs with subheadings to organize the paper better. It is currently very difficult to read and be engaged. 

Response: Thank you very much for your suggestion. We agree with your proposal and we have restructured our article and added subheadings as well. We hope our article will be easier to read after the revision.

Comment 6: Many of the figures, especially forest maps, are so small/pixelated and impossible to read.

Response: Thank you very much for your discovery, we can provide the original image of the article in PNG, JPEG format. We believe the figures and text of the original images are clear and readable.

Reviewer 5 Report

In the presented review, Xia et al summarized the recent advances in clinical trials of existing factor XI inhibitors the perspective to use these inhibitors as anticoagulant therapy. The review is well written and represent all novel literature in the field. I have only small suggestions to the authors that will improve this nice review.

1.      It is important to include small summary after each chapter (ASO, small molecular, better add inhibitors, antibody). The authors compared different inhibitors in each classes, but no summary and their opinion on differences between these inhibitors in each class.

2.      Quality of Figs 3, 6, and 7 should be improved. It is difficult to read, the font is very small and I will suggest to use bold that it will be easy to read.

3.      Some small improvements of sentences:

P. 5, L. 181. effects. there are. Delete point.

P. 5, L. 195. VKA, Because. because

P. 5, L. 205. an ischemic stroke (IS) and deep vein thrombosis (DVT). Abbreviations were introduced previously.

P. 10, L. 469. B Alexander Yi et. B?

P 11, L. 487. by Thomas et al. Better to put ref.83 here.

P. 11, L. 494. Researchers investigated examined?

P. 11, L. 515. high (210mg) doses. (NCT04523220). Delete point.

P. 11, L. 522. Lorentz et al. Better to put ref.86 here.

Author Response

Comment 1:  It is important to include small summary after each chapter (ASO, small molecular, better add inhibitors, antibody). The authors compared different inhibitors in each classes, but no summary and their opinion on differences between these inhibitors in each class.

Response: Thank you very much for your suggestion and also for your approval of our article. Your suggestion is a study that is well worth discussing in depth and we hope that, in the future, when more clinical trials are published to make a reticulate meta of the efficacy and safety of the three classes of drugs, the differences between each class of inhibitors will be described with objective data. We think this could be a good direction for research. Thank you for providing us with your thoughts, we are very inspired.

Comment 2:   Quality of Figs 3, 6, and 7 should be improved. It is difficult to read, the font is very small and I will suggest to use bold that it will be easy to read.

Response: Thank you very much for your discovery, we can provide the original image of the article in PNG, JPEG format. We believe the figures and text of the original images are clear and readable.

Comment 3: Some small improvements of sentences:

  1. 5, L. 181. effects. there are. Delete point.
  2. 5, L. 195. VKA, Because. Because
  3. 5, L. 205. an ischemic stroke (IS) and deep vein thrombosis (DVT). Abbreviations were introduced previously.
  4. 10, L. 469. B Alexander Yi et. B?

P 11, L. 487. by Thomas et al. Better to put ref.83 here.

  1. 11, L. 494. Researchers investigated examined?
  2. 11, L. 515. high (210mg) doses. (NCT04523220). Delete point.
  3. 11, L. 522. Lorentz et al. Better to put ref.86 here.

Response: Thank you very much for your attention to the details of our manuscript, which we have revised one by one based on your suggestions. Thank you again for your conscientiousness and rigor.

Round 2

Reviewer 1 Report

Thank you for including me in the review of the revised article entitled: ‘Factor XIa inhibitors as a novel anticoagulation target: recent clinical research advances’. Indeed, the manuscript improved considerably since its first version. However, several points of the review were not implemented.

-        The text is superfluous. I proposed a reduction to 3500-5000 word. The discussion is overwhelmed by description of the individual trials without any apparent synthesis but repetition of already published data.

-        The abstract contains no numerical data of the results. Generic text (like descriptions of heterogeneity measures, electronic search and data extraction) should be replaced with Results.

-        The text have an undulating niveau of medical language: See for example of the opening statement of the abstract:“Although anticoagulant drugs used clinically have achieved some efficacy, there are many problems with these drugs, which cause increasing hazards of serious bleeding in patients, particularly hemorrhage of the digestive tract, intracranial bleeding, and life-threatening major hemorrhages” a thorught overview is necessary rephrasing like “While current clinically-administered anticoagulant medications have demonstrated effectiveness, they also pose significant risks. These drugs can precipitate severe bleeding complications in patients, including, but not limited to, gastrointestinal hemorrhages, intracranial bleeding, and other life-threatening major bleedings.” aso

-        Methods: correct the following statements: “The effectiveness and safety of FXIa inhibitors were assessed by a distinct quantitative analysis of clinical studies. The Mantel-Haenszel (Fixed effects approach for meta-analysis) with a random effects model was utilized to evaluate pooled effect sizes for meta-analyses.”,  “(Factor Eleven An Inhibitors))” Meaningful description of the metaregression plot is missing.

-        Figure 2. 5. and 7. The trial of osocimab is included twice into the analysis: in fact this was a three arm trial that of duplicating the active arm may bias the analysis (giving an extra ~10% weight to the data of this 441 patients) either arm should be excluded or if the information graph allows network analysis should be presented.

-        Figure 4. And 9  Is merely a subgroup of analysis of the earlier figure 2 and 4. One of the two should be deleted.

Author Response

Comment 1: The text is superfluous. I proposed a reduction to 3500-5000 word. The discussion is overwhelmed by description of the individual trials without any apparent synthesis but repetition of already published data.

Response: Thank you very much for your suggestion. We have tried our best to trim down the article to about 5700 words excluding the abstract. If there are any problems, we can still continue to modify.

Comment 2: The abstract contains no numerical data of the results. Generic text (like descriptions of heterogeneity measures, electronic search and data extraction) should be replaced with Results.

Response: Thank you very much for your suggestion and we have revised the abstract.

Comment 3: The text have an undulating niveau of medical language: See for example of the opening statement of the abstract:“Although anticoagulant drugs used clinically have achieved some efficacy, there are many problems with these drugs, which cause increasing hazards of serious bleeding in patients, particularly hemorrhage of the digestive tract, intracranial bleeding, and life-threatening major hemorrhages” a thorught overview is necessary rephrasing like “While current clinically-administered anticoagulant medications have demonstrated effectiveness, they also pose significant risks. These drugs can precipitate severe bleeding complications in patients, including, but not limited to, gastrointestinal hemorrhages, intracranial bleeding, and other life-threatening major bleedings.” aso

Response: Thank you very much for your suggestion, we have modified the suggestion.

Comment 4: Methods: correct the following statements: “The effectiveness and safety of FXIa inhibitors were assessed by a distinct quantitative analysis of clinical studies. The Mantel-Haenszel (Fixed effects approach for meta-analysis) with a random effects model was utilized to evaluate pooled effect sizes for meta-analyses.”,  “(Factor Eleven An Inhibitors))” Meaningful description of the metaregression plot is missing.

Response: Thank you very much for your suggestion, we have modified the suggestion.

Comment 5: Figure 2. 5. and 7. The trial of osocimab is included twice into the analysis: in fact this was a three arm trial that of duplicating the active arm may bias the analysis (giving an extra ~10% weight to the data of this 441 patients) either arm should be excluded or if the information graph allows network analysis should be presented.

Response: Thank you very much for your suggestion and we agree with you. Because there were very few clinical trials evaluating the safety and efficacy of FXIa factor inhibitors in the studies we included, we chose to split them into two trials for this reason. We think your suggestion is reasonable and we will take into account the error caused by the increased weights when we update our meta-analysis in the future. Thank you very much for your professional and rigorous approach.

Comment 6: Figure 4. And 9  Is merely a subgroup of analysis of the earlier figure 2 and 4. One of the two should be deleted.

Response: Thank you very much for your suggestion, we have removed Figure 4.

Reviewer 4 Report

Thank you for addressing all of my comments!

There are still some very minor spacing/spelling issues that can be addressed during proof process

Author Response

Comment 1: There are still some very minor spacing/spelling issues that can be addressed during proof process

Response: Thank you very much for your suggestion, we read the whole text carefully and made changes where we could find them. If there are any other issues, we can make final corrections at the proof stage. Thank you very much for your suggestion. Best wishes to you.